# Highly Sensitive Fluorescent Detection of Acetylcholine Based on the Enhanced Peroxidase-Like Activity of Histidine Coated Magnetic Nanoparticles

**DOI:** 10.3390/nano11051207

**Published:** 2021-05-01

**Authors:** Hong Jae Cheon, Quynh Huong Nguyen, Moon Il Kim

**Affiliations:** Department of BioNano Technology, Gachon University, 1342 Seongnamdae-ro, Sujeong-gu, Seongnam 13120, Gyeonggi, Korea; hjchun1201@naver.com (H.J.C.); pruedence122@gmail.com (Q.H.N.)

**Keywords:** histidine coated magnetic nanoparticles, peroxidase mimics, nanozyme, acetylcholine detection, Alzheimer’s disease

## Abstract

Inspired by the active site structure of natural horseradish peroxidase having iron as a pivotal element with coordinated histidine residues, we have developed histidine coated magnetic nanoparticles (His@MNPs) with relatively uniform and small sizes (less than 10 nm) through one-pot heat treatment. In comparison to pristine MNPs and other amino acid coated MNPs, His@MNPs exhibited a considerably enhanced peroxidase-imitating activity, approaching 10-fold higher in catalytic reactions. With the high activity, His@MNPs then were exploited to detect the important neurotransmitter acetylcholine. By coupling choline oxidase and acetylcholine esterase with His@MNPs as peroxidase mimics, target choline and acetylcholine were successfully detected via fluorescent mode with high specificity and sensitivity with the limits of detection down to 200 and 100 nM, respectively. The diagnostic capability of the method is demonstrated by analyzing acetylcholine in human blood serum. This study thus demonstrates the potential of utilizing His@MNPs as peroxidase-mimicking nanozymes for detecting important biological and clinical targets with high sensitivity and reliability.

## 1. Introduction

Acetylcholine (ACh) is a vitally important neurotransmitter that is closely related to functions of both the peripheral and central nervous systems that control various human physiological processes and behaviors. Abnormal amounts of ACh, thus, are able to provoke several cognitive or neural disorders, consisting of progressing dementia, schizophrenia, and Parkinson’s and Alzheimer’s diseases [1,2,3,4]. Alzheimer’s disease, in fact, belongs to the top 10 leading causes of death in the United State and is considered as one of the major healthcare and economic burdens across the globe [5,6]. In order to determine the level of ACh in clinical samples, several conventional methods such as mass spectrometry, gas chromatography, and high performance liquid chromatography have been exploited [2,7,8]. Though these approaches are sensitive and reliable, they are generally expensive and somewhat burdensome to perform, as they require sample pretreatment and sophisticated instrumentations as well as skilled technicians, which can only be found in centralized hospitals and laboratories [2,7,8]. The prolonged time for diagnosis is also a hurdle that prevents these methods from being broadly utilized in regular testing of ACh in clinical samples. Therefore, it is crucial to develop a simple, sensitive, but reliable approach for frequent ACh level examination that can not only help detect and diagnose the disease at premature stages but also have better clinical preventions and treatments [9].

Nanomaterials that can perform enzyme-like activities are defined as nanozymes and considered to be the rival for natural enzymes in aspects such as high stability, low cost, and the robustness to be stored for a relatively long time [10]. Since the first peroxidase-mimicking nanozymes, magnetic nanoparticles (MNPs), were discovered in 2007 [11], plenty of nanozymes have been developed and studied [10,12,13,14,15,16]. However, MNPs were still considered as fascinating materials in numerous biosensing fields [17,18,19,20,21] and biomedical applications such as drug delivery, magnetic hyperthermia, and biomedical imaging, owing to their inherent biocompatibility, low toxicity, as well as their exotic response to the magnetic field [22,23]. Though the peroxidase-mimicking MNPs are more stable in a broad range of pH and temperature values than natural horseradish peroxidase (HRP), their practical applications in the sensing field are somewhat limited due to their relatively low catalytic activity and specificity [18]. Thus, further modifications for size and morphology optimization, or surface coating to ameliorate enzymatic activity of MNPs with a simple synthetic procedure, are still an attractive topic in this research field. In fact, there have been many attempts to modify the surface of MNPs with other molecules, including SiO_2_, 3-aminopropyl triethoxysilane (APTES), dextran, and poly (ethylene glycol) (PEG). Unfortunately, these alterations could lead to surface shield effects that prevent catalytic sites of MNPs from substrates, resulting in lower catalytic performances [11]. Other studies utilized active polymers such as chitosan to protect MNPs from oxidation as well as provide active amino groups for subsequent immobilizations or attachments instead of boosting catalytic activity [24,25]. Therefore, it is necessary to develop other efficient strategies to modify MNPs that perform better peroxidase activity.

Motivated by the necessity to enhance the catalytic activity of MNPs, we have developed histidine coated magnetic nanoparticles (His@MNPs), inspired by the natural architecture of the active site of HRP, which contains a heme group having iron as central to its catalytic molecules and two coordinated histidine residues. The as-synthesized His@MNPs possessed uniform spherical morphology with application-friendly sizes of less than 10 nm and exhibited much higher peroxidase activity compared to pristine or other amino acids coated MNPs. With the amplified catalytic activity, the newly developed His@MNPs were applied to detect ACh and choline levels in clinical samples. The simple synthesis and effortless detection procedure of this system shows a promise for robust and trustworthy detection of ACh with reduced costs and intricacies.

## 2. Materials and Methods

### 2.1. Materials

Iron (II) chloride tetrahydrate, L-alanine (Ala), L-arginine (Arg), L-histidine (His), L-lysine (Lys), L-methionine (Met), sodium hydroxide, choline chloride, acetylcholine chloride (ACh), choline oxidase (ChOx), acetylcholinesterase (AChE) from human erythrocytes, human serum, 3,3′,5,5′-tetramethylbenzidine (TMB), lactose, aspartic acid, glycine, glutamic acid, and glutathione were purchased from Sigma-Aldrich (Milwaukee, WI). Hydrogen peroxide (30%, H_2_O_2_) was purchased from Junsei Chemical Co. (Tokyo, Japan). Amplex UltraRed (AUR) reagent was purchased from Thermo Fisher Scientific (Waltham, MA, USA). Choline/Acetylcholine assay kit (ab6534) was purchased from Abcam company (Cambridge, MA, USA). All other chemicals were of analytical grade or higher, and used without further purification. All solutions were prepared with DI water purified by a Milli-Q Purification system (Millipore Sigma, Burlington, MA, USA).

### 2.2. One-Pot Synthesis of Histidine and other Amino Acids Coated Magnetic Nanoparticles

His@MNPs was synthesized as follows. Briefly, 0.3 g of His was added into 50 mL of 75.84 mM aqueous FeCl_2_.4H_2_O solution, followed by vigorous stirring for 12 h at 25 °C. Subsequently, the above mixture was added dropwise to 50 mL of 200 mM sodium hydroxide solution with continuous stirring and incubated during the course of 30 min for reaction. Afterwards, the black precipitates were collected by the magnet, followed by several washing steps with ethanol and distilled water. Finally, the resulting His@MNPs were dried in a vacuum oven at 60 °C. As controls, other amino acids coated MNPs, including Met, Arg, Ala, and Lys coated MNPs, were also prepared as aforementioned procedures except the kinds of employed amino acid.

### 2.3. Characterization of His@MNPs

The as-prepared His@MNPs were characterized by using scanning transmission electron microscopy (STEM). To examine the element compositions, the energy-dispersive spectrometer (EDS) (Bruker, Billerica, MA, USA) was employed. Fourier transform infrared (FT-IR) spectra of His@MNPs were obtained using FT-IR spectrophotometer (FT-IR-4600, JASCO, Easton, MD, USA). The crystal structure of nanomaterials was determined by utilizing X-ray diffraction (Rigaku Corporation, Tokyo, Japan).

### 2.4. Investigation for the Peroxidase-Like Activity of His@MNPs and Controls

The peroxidase-mimicking activity of His@MNPs was demonstrated via the oxidation of TMB in the presence of H_2_O_2_. Generally, 100 µg/mL of His@MNPs was added into sodium acetate buffer (0.1 M, pH 4.0) containing 100 mM H_2_O_2_ and 5 mM TMB, followed by 10 min incubation at 40 °C. After the reaction, the color change from colorless to blue of oxidized TMB can be observed by the naked eye, and data were recorded in scanning mode by employing a microplate reader (Synergy H1, BioTek, Winooski, VT, USA). The oxidation of controls, including pristine MNPs as well as other amino acids coated MNPs (Met@MNPs, Arg@MNPs, Ala@MNPs and Lys@MNPs), were also carried out and recorded for comparisons.

The peroxidase-imitating activities of His@MNPs and pristine MNPs were further elucidated via steady-state kinetic analysis deploying TMB and H_2_O_2_ as substrates. The investigations were designed by varying the concentration of one substrate concentration while fixing another substrate at a saturated concentration. For the kinetic assay of TMB, 1 M of H_2_O_2_ was added to 1 mL of reaction buffer with varying concentrations of TMB from 3 µM to 800 µM. For H_2_O_2_-dependent kinetic assay, 800 µM of TMB was utilized with varying concentrations of H_2_O_2_ from 0.03 M to 1 M. Kinetic parameters including the apparent Michaelis-Menten constant (K_m_) and maximum initial velocity (V_max_) were computed using the Lineweaver-Burk plots and Michaelis-Menten equation as follows: ν = V_max_ × 1/(K_m_ + [S]), where ν is the initial velocity and [S] is the concentration of substrate [3,11]. The catalytic constant k_cat_ was stemmed from k_cat_ = V_max_/[E], where [E] is the iron concentration quantified by inductively coupled plasma atomic emission spectroscopy (ICP-AES, Polyscan 60E, Thermo Jarrell Ash, Franklin, MA, USA) method.

### 2.5. Detection of H_2_O_2_, Choline, and ACh Using His@MNPs

Fluorescent detection of H_2_O_2_, choline, and ACh was conducted by utilizing AUR as a substrate in a black 96-well plate, and the data were recorded by using a microplate reader. For the detection of choline and ACh, Tris-acetate buffer at pH 7.0 was utilized due to the pH-dependent nature of ChOx and AChE, which show their activities at around neutral pH [26,27]. Regarding H_2_O_2_ detection, a total 200 µL solution consisting of 100 µg/mL His@MNPs, 10 µM AUR, and H_2_O_2_ at various concentrations (0–250 mM) was incubated at 40 °C for 15 min. The resulting fluorescence signals of oxidized AUR were recorded by a microplate reader with the excitation and emission wavelength at 490 nm and 590 nm, respectively. In terms of choline determination, the experiment was conducted via the following procedure. A total 200 µL of mixture reaction containing His@MNP (100 µg/mL), ChOx (0.6 U/mL), AUR (10 µM), and different concentrations of choline (0–200 µM) in Tris-acetate buffer (10 mM, pH 7.0) was incubated at 40 °C for 15 min. The measurement was performed by following the same aforementioned procedures for detection of H_2_O_2_. ACh detection by His@MNPs was carried out by incubating His@MNP (100 µg/mL), AChE (5 U/mL), ChOx (0.6 U/mL), AUR (10 µM), and ACh with different concentrations (0–250 µM) in the Tris-acetate buffer for 15 min at 40 °C. The signal results were recorded by the same procedure described above.

For the selectivity assay toward ACh, several potentially interfering biological compounds, such as carbohydrates (glucose, lactose), amino acids (aspartic acid, glycine, and glutamic acid), and biothiol (glutathione), were employed with 10-times higher concentration (100 µM) in comparison with target ACh (10 µM).

The concentrations of ACh in spiked human blood serum samples were determined through the standard addition method. Prior to addition, the human serum was diluted to 10% and predetermined amounts of ACh (3, 5, and 8 µM) were added to make spiked samples. Accordingly, the recovery rate [recovery (%) = measured value/added value × 100] and the coefficient of variation [CV (%) = SD/average × 100] were computed as the described equations to assess the precision and reproducibility of the assays.

## 3. Results and Discussion

### 3.1. Synthesis and Characterization of His@MNPs

A simple but efficient one-pot heat-treatment method was developed to synthesize His@MNPs that exhibit significantly amplified peroxidase-like activity and were utilized to detect ACh. The detection regime was based on the consecutive catalytic reactions triggered from ACh, with the involvement of multiple enzymes including AChE and ChOx, to create H_2_O_2_ as a final, direct product for peroxidase reaction. The peroxidase reaction was performed by His@MNPs, which oxidize the AUR substrate into a highly fluorescent product (AURox) (Scheme 1). The prepared materials then were well characterized by different techniques. Firstly, the morphology and size of His@MNPs and pristine MNPs were examined via TEM analysis, as shown in Figure 1a,b. The TEM images revealed that both pristine MNPs and His@MNPs were well prepared with uniform, spherical shapes and sizes of less than 10 nm, which are desirable for catalytic applications. The size of His@MNPs was larger than that of pristine MNPs (roughly 2–3 nm larger), which can plausibly be attributed to the histidine coating layer. To validate that, EDS analysis was deployed to identify the contributing elements of His@MNPs. From Figure 1c, it can be observed that all the components are well distributed. The occurrence of a nitrogen element in His@MNPs instead of unmodified MNPs proved that the histidine layer was well deposited. Additionally, the difference in zeta potential between His@MNPs and MNPs could also be deduced from the presence of a histidine layer on the surface of MNPs (Appendix A). The signature XRD patterns of pristine MNPs are shown in Appendix Aa, which was well aligned with the standard JCPDS database (JCPDS 00-019-0629). These peaks are strong and distinct, implying the good crystallinity of bare MNPs. In respect to His@MNPs, though the peaks pattern is analogous to that of pristine MNP, the intensities are apparently decreased, which indicates the presence of a histidine coating layer. In the FT-IR spectra (Appendix Ab), the characteristic peaks at 540 cm^−1^ and at around 3300–3400 cm^−1^, which derived from Fe-O bond and O-H stretching vibration, respectively, were observed in both MNPs and His@MNPs, proving the occurrence of iron oxide particles [28]. Furthermore, the appearance of a peak at 1120 and 1386 cm^−1^ represented C-O stretching and COO- group originating from histidine residues. The broad peak at 1630 cm^−1^ in the His@MNPs pattern also indicated the presence of C = O stretching frequency which derived from histidine [29]. According to these results, the coating of histidine on the surface of MNPs was consolidated.

### 3.2. Enhanced Peroxidase-Like Activity of His@MNPs

The catalytic activity of His@MNPs was investigated via the peroxidase-facilitated oxidation of colorimetric reagent TMB as substrate in the presence of H_2_O_2_. Due to the peroxidase-like effect of His@MNPs, reactive oxygen species (ROS) were created which further react with TMB to produce oxidized TMB, and thus the color of the reaction solution changed from colorless to blue. For comparison, the peroxidase-imitating activities of other amino acids coated MNPs, including Met@MNPs, Arg@MNPs, Ala@MNPs, and Lys@MNPs, as well as the pristine MNPs, were also examined. According to the results, the catalytic activity of His@MNPs was 10-fold higher compared to bare MNPs and was significantly higher than other amino acids coated MNPs (Figure 2). This result was also equivalent to the previously reported study [30], which describes the importance of mimicking the natural active site of HRP. Ferric and ferrous ions present on the His@MNPs might catalyze the peroxidase reaction, and the histidine residues additionally contributed to the catalysis, like the natural active site of HRP. Preliminary optimizations related to temperature and pH effects on the peroxidase-like activity of His@MNPs were also conducted. The results, henceforth, revealed that pH 4.0 and temperature at 40~50 °C were optimum conditions for the catalytic activity of His@MNPs (Appendix A).

To further understand the affinity between His@MNPs and substrates (TMB and H_2_O_2_), steady-state kinetic values including K_m_, V_max_, and k_cat_ were computed and compared with those obtained from pristine MNPs. Figure 3 showed the typical Michaelis-Menten curves of initial velocities against various concentrations of TMB and H_2_O_2_ which were further used to draw Lineweaver-Burk plots. The Lineweaver-Burk plots then were applied to determine K_m_ and V_max_. Table 1 summarized the catalytic parameters of His@MNPs, unmodified pristine MNPs, HRP, and other peroxidase-like nanozymes which were recently reported [2,3,11,16,31]. According to the Table 1, it is noticeable that the K_m_ value (TMB) for His@MNPs was approximately 3 times higher than that of natural HRP. However, the K_m_ value of His@MNPs toward H_2_O_2_ was roughly half that of bare MNPs, indicating that the histidine modification efficiently increased the affinity toward H_2_O_2_ rather than TMB, as similarly reported previously [30]. Though His@MNP showed marginally lower affinity toward TMB, its catalytic efficiency (k_cat_) was determined to be higher than those obtained from other nanozymes and HRP. The high k_cat_ value could be attributed to the presence of many efficient active sites by the histidine modifications on MNPs.

Unlike natural enzymes, which are susceptible to harsh conditions, nanozymes are highly stable, withstanding even high temperatures or extreme pH conditions, in which most enzymes are denatured and lose their activities. To demonstrate the robustness of His@MNPs, various pH and temperature conditions were applied and the results were depicted in Appendix A. In particular, His@MNPs conserved their activities at various pH conditions, even at acidic or basic conditions (where it remained over 90% active), whereas HRP lost roughly 60% activity within basic pH conditions. Likewise, HRP began to plunge when the temperature was higher than 30 °C, losing approximately 70% of activity at 37 °C, while His@MNPs still retained acceptable activity over 70%. The minor loss of activity in His@MNPs at high temperature could conceivably be explained due to the histidine organic component. However, the activity loss is marginal, proving the high thermal resistance of the as-developed His@MNPs.

In general, the peroxidase-mimicking activity is demonstrated by confirming the generation of free hydroxyl (·OH) radicals from H_2_O_2_ [32,33]. To test this, a non-fluorescent TA probe was deployed to examine the formation of free OH radicals, which are produced during the decomposition of H_2_O_2_. Upon the formation of OH, the non-fluorescent TA molecules transform into highly fluorescent products (2-hydroxy terephthalic acid), which emit a unique fluorescence signal at around 435 nm. It is worth mentioning that, compared to bare MNPs, the amount of free OH radicals generated from His@MNPs was considerably increased as demonstrated in Figure 4, indicating their higher catalytic activity.

### 3.3. Highly Sensitive Fluorescent Detection of Choline and ACh

The level of ACh in clinical samples was determined via cascade catalytic reactions starting from AChE coupled with ChOx, followed by a peroxidase-facilitated reaction performed by His@MNPs as follows.
(1)$$Acetylcholine\;\overset{AChE}{\to }\;Choline+acetic\;acid$$
(2)$$Choline\;\overset{ChOx}{\to }\;betaine+H_{2}O_{2}$$
(3)$$H_{2}O_{2}+AUR\overset{His@MNP}{\to }H_{2}O+AURox$$

According to the above chain reactions, it can be observed that in the occurrence of ACh, enzyme AChE catalyzes its hydrolysis to generate choline, which is the specific substrate for choline oxidase. After choline molecules were cleaved by ChOx, H_2_O_2_ molecules were formed and subsequently consumed by the peroxidase-mediated reaction catalyzed by His@MNPs. This reaction produces free ·OH radicals that interact with AUR to create oxidized AUR. Based on previous reports, the concentration of ACh in body fluid is extremely low, generally in the nanomolar range [34,35]. Thus, AUR was exploited instead of other peroxidase substrates to monitor the presence of ACh since it can produce a very sensitive, highly fluorescent product (AURox). By recruiting AUR as a peroxidase substrate, the enzymatic reaction catalyzed by His@MNPs was performed at neutral pH (pH 7.0).

Under the aforementioned conditions, the peroxidase activity of His@MNPs to detect H_2_O_2_ with AUR as a substrate was examined. The results, which were shown in Appendix A, demonstrated that within a suitable range, the fluorescence signals derived from oxidized AUR increased as the concentrations of H_2_O_2_ raised, which established a good linear correlation (R^2^ > 0.99). Accordingly, we further investigated the sensitivity of this system towards choline and ACh. In detail, as the concentration of choline increases from 0.3 µM to 5.0 µM, a positively proportional increase was observed in the respective fluorescence intensity, thus constructing a highly linear correlation (R^2^ > 0.99) (Figure 5a and Appendix A). Calculated from the linear regression equation (y = 0.9449x + 294.63), the limit of detection (LOD) was determined to be 200 nM. The LOD value was calculated based on the formula: LOD = 3 × δ/slope, where δ is the standard deviation of blank and slope is the slope of calibration curve. Subsequently, a convenient, one-step operation was carried out to detect ACh via the multiple enzymatic reactions (AChE, ChOx, and peroxidase-mimicking His@MNPs). Figure 5b as well as Appendix A showed a qualified linear relationship (R^2^ > 0.99) between the fluorescent signal and concentration of ACh (0.25 µM to 5 µM). Particularly, a linear equation was established (y = 5.9983x + 539.76) when the concentration of ACh increases from 0.25 µM to 5 µM. Based on this equation, LOD down to 100 nM was achieved, which was lower than those of recent studies [3,8,36].

For biosensors, selectivity is an important criterion to assess the practical property, especially in the diagnostic area [3]. To confirm the selectivity of the developed His@MNPs based sensing regime, several interfering substances were utilized, including glucose, lactose, aspartic acid, glycine, glutamic acid and glutathione. The results illustrated in Figure 5c indicated that only in the presence of Ach can a significant fluorescent signal be undoubtedly recognized, while other substances did not generate any considerable signals, despite the fact that their concentrations were used at 10-times higher compared to the target ACh.

The practical application of the developed regime was further verified by detecting the level of ACh in human serum (10%) via a standard addition method. The obtained results exhibited an excellent precision, yielding CVs in the range of 1.16 to 7.54% and recovery rates from 100.20% to 101.92%, demonstrating a good agreement with the spiked amount of ACh (Table 2). Thus, the results confirm that the developed sensing systems utilizing multiple enzymes and His@MNPs as peroxidase mimics are promising to be applied for the real detection of ACh in clinical settings. Additionally, our His@MNPs based method was compared with a commercial Choline/Acetylcholine Assay Kit (ab65345) for quantitatively determining target acetylcholine. The observed correlation between the two methods was R^2^ > 0.99 (Appendix A). This high concordance between the two methods ensured the validity of the His@MNPs based strategy for reliable quantitative determination of acetylcholine.

## 4. Conclusions

In conclusion, we have successfully synthesized very small-sized His@MNPs via a facile one-pot method, yielding their significant enhancement in the peroxidase-like activity. Compared to pristine MNPs, the newly synthesized His@MNPs, which were designed to mimic the structure of active site of natural HRP, exhibited approximately 10-fold higher peroxidase-like activity. The activity enhancement was mainly due to the increased affinity toward H_2_O_2_ and the presence of many efficient active sites by the histidine modifications on MNPs. By coupling with appropriate enzymes including AChE and ChOx, the developed nanozymes then were successfully exploited to detect ACh with high specificity and sensitivity in which the LOD was recorded to be as low as 100 nM. With the satisfied outcomes, our developed nanozymes with good stability and enhanced peroxidase-mimicking activity are sufficient to be utilized for biological sensing applications.

## Data Availability

Not applicable.

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
