# Peer review of "Highly Sensitive Fluorescent Detection of Acetylcholine Based on the Enhanced Peroxidase-Like Activity of Histidine Coated Magnetic Nanoparticles"

_nanomaterials, 2021, doi:10.3390/nano11051207_

Round 1

Reviewer 1 Report

In this manuscript, the authors described the design and synthesis of histidine-coated magnetic nanoparticles (His@MNPs) as peroxidase-mimicking nanozymes for detection of choline and acetylcholine. His@MNPs shows high selectivity to acetylcholine than other amino acids. These peroxidase-mimicking nanozymes could be successfully applied to detect the acetylcholine in the human serum samples. Overall, the manuscript presents well and the data in the manuscript support the result basically. Moreover, it provides a general strategy for the development of nanozyme. Thus, I suggest it can be considered acceptable after addressing the following issues.

  1. What is the underline mechanism of His@MNPs as peroxidase-mimicking nanozymes? What is the function of MNPs in this system or it just acts as the nanocarriers?
  2. Compared with other peroxidase-mimicking nanozymes, as shown in Table 1, His@MNPs present the highest catalytic efficiency but the poor affinity to substrates (high Km). Please give an explanation for this.
  3. Table 2 shows that the author has tested human serum samples, and it is recommended to compare the results with commercial kits.
  4. Figure issues: Figures should be aligned well and provided with higher resolution. For instance, in figure 3, c) and d) are not aligned. 

Reviewer 2 Report

Comments:

   This paper synthesizes an enhanced peroxidase-imitating activity nanomaterial (His@MNPs) inspired by the active site structure of HRP, and systematically studied its peroxidase activity. Then based on the peroxidase-imitating activity of the nanomaterial, a fluorescent detection method for choline and acetylcholine assay is develops. The research has a certain degree of innovation, and can be published in this journal after modification.

Some specific comments are as follows:

  1. The title and abstract omit some important information, such as the output mode of sensor signal.
  2. In the experimental method, some important information seems to be omitted by the author, such as the specific treatment process of the sample.
  3. The feasibility analysis of this study is missing.
  4. The information of sensitivity emission spectrum is missing.
  5. Sensitivity, specificity and actual sample analysis are written together, which makes it more difficult for readers to obtain relevant information.
  6. The conclusion seems to be too simple, and it seems to omit the innovative information of this paper.
  7. There exist some grammatical errors and mistakes. English editing should be required for further processes.
  8. Some Figures are hard to read. These issues should be solved.

Reviewer 3 Report

In their manuscript "Highly sensitive acetylcholine detection based on the enhanced peroxidase-like activity of histidine coated magnetic nanoparticles", the authors described the use of a peroxidase-imitating activity, coupled with choline oxidase and acetylcholine esterase, for the detection of choline and acetylcholine in blood serum. The increased efficiency of the histidine coated nanoparticles was already showed by Fan et al., [reference 26 of manuscript], authors extend these results to an application for choline and acetylcholine indirect detection. However, authors lack in giving some information necessary to support their findings, so I suggest reconsidering after major revisions.

Major revisions

Rows 33-36. Authors have to change this part of the introduction, because could generate a misunderstanding about the possibility to monitor or diagnostic disease like Alzheimer or Parkinson, measuring the acetylcholine in the blood. There was no correlation between the blood acetylcholine level and age, sex or disease, just people who have myasthenia gravis (a rare autoimmune disease) could be affected by the presence of abnormal protein (acetylcholine receptor antibody) interferings with acetylcholine, but in this case the test measure the presence of the antibody.

Row 37. What “regular diagnosis” are the authors referring to?

Row 41. What “routine medical check-ups” are the authors referring to?

Rows 42-43. I’m not completely in agreement with the sentence: “... for frequent ACh level examination that can not only help diagnose the disease at premature stages but also have better clinical preventions and treatments.”, could author support this sentence with references?

Rows 107, 139, 145, 209, authors use different pH for the standard condition (4.0) and the assay of choline/acetylcholine (7.0). To me it is clear that in presence of the enzymes (ChOx and AChE) the pH 7.0 is required, but I suggest explaining for readers not in the field.

Row 115. Please indicate the range of concentrations for TMB and H2O2. Also describe how the measurements are performed, including the saturated concentration of TMB and H2O2 used.

Row 124. This paragraph could move to supplementary materials.

Row 130. Please indicate the pristine MNPs and His@MNPs range of concentrations.

Row 139. Please indicate the range of H2O2 concentrations.

Row 144. Please indicate the range of choline concentrations.

Row 148. Please indicate the range of ACh concentrations.

Rows 151-155. Authors well described how calculate the amount of choline and ACh in human serum samples, but they have also to describe how the experiments were performed. In fact, it is not clear if choline, or ACh, was added to the serum or to the assay solution containing the serum, also the amount of serum used is not described (probably 10% as indicate in results?).

Row 158. Remove scheme 1a, because not necessary, instead improve the scheme 1b making more clear the enzymes used, the products of reaction, and the oxido-reduction process of H2O2/AUR using His@MNP, such as scheme 1 in reference 2 (Gou et al.) or rows 263-265.

Row 184. In absence of a scale in supplementary figure 2a, how the authors observe an apparent decrease in the intensities?

Row 225 and Table 1. Authors should report also data from other references, even if better with respect their data, such as reference 2.

Rows 246-254. Authors have to describe something about the mechanism of the enzyme-mimicking activity of His@MNP, such as its catalase activity, or explain the usefulness of these data. Otherwise, this paragraph is not necessary, as well as the paragraph 2.5.

Row 271. The blood acetylcholine levels fall into the range of 0.20 to 1.31 microM, it is not clear why the authors explore range a hundred times higher.

Figure 5 and S5. It is not clear if authors remove the background from their data (or normalize), also curve are not linear. Moreover, it is not clear how a lower choline concentration will result in a higher signal (fig. 5a)  with respect to a thousand time concentration of H2O2 (fig. S5), which is a secondary product of choline oxidation.

Row 285. The LOD calculation is not clear, can authors explain?

Row 293. Nothing is described about these measurements in the material and methods, also authors have to explain the reasons they used these compounds.

Row 305. Authors have to explain the assay in materials and methods, because it appears that they add acetylcholine to the assay solution and not to the serum, so being the serum diluted 10 times (?), this means that the in serum concentration of acetylcholine detectable by their method will be 10 times higher (30 microM), very far from the real blood concentration.

Row. 326. How can this activity applied to the environmental monitoring? Can authors explain this part? or remove?

Minor comments

Row 10. Remove “(HRP)” from the abstract.

Row 106. Change “existence” in “presence”.

Row 106. Change “existence” in “presence”.

Add the “measure unit” on y-axis of figures, also supplementary ones, probably a.u. (arbitrary units).

Round 2

Reviewer 2 Report

The author makes a detailed and serious improvement on the suggestions put forward by the reviewers, and basically meets the requirements of publication. It is suggested that the manuscript be accepted.

Author Response

Dr. Moon Il Kim

April 27, 2021

Dear Editor,

Enclosed, please find our revised manuscript entitled “Highly sensitive fluorescent detection of acetylcholine based on the enhanced peroxidase-like activity of histidine coated magnetic nanoparticles” (nanomaterials-1188677), which we would like considered for publication as an article in Nanomaterials. We appreciate you for the opportunity you gave us to revise the manuscript, and are grateful for the positive evaluation and valuable suggestions of the reviewers.

Responses to Reviewers’ Comments

Reviewer #2

The author makes a detailed and serious improvement on the suggestions put forward by the reviewers, and basically meets the requirements of publication. It is suggested that the manuscript be accepted.

We believe that we have satisfactorily responded to the questions/comments made by the reviewers. The excellent suggestions made by the reviewers have enabled us to create a much more informative and clearer manuscript. Please do not hesitate to contact me if you have any questions. Thank you very much for considering this manuscript for publication in ‘Nanomaterials’. We look forward to hearing from you soon.

Sincerely yours,

Moon Il Kim, Ph.D.